# Coix Seed Consumption Affects the Gut Microbiota and the Peripheral Lymphocyte Subset Profiles of Healthy Male Adults

**DOI:** 10.3390/nu13114079

**Published:** 2021-11-15

**Authors:** Minami Jinnouchi, Taisei Miyahara, Yoshio Suzuki

**Affiliations:** Graduate School of Health and Sports Science, Juntendo University, Inzai 270-1695, Chiba, Japan; j.minami.67@gmail.com (M.J.); tl_bpnpne_22@yahoo.co.jp (T.M.)

**Keywords:** adlay, pearl barley, gut microbiota, immune system, herbal medicine

## Abstract

A systematic examination of the effects of traditional herbal medicines including their mechanisms could allow for their effective use and provide opportunities to develop new medicines. Coix seed has been suggested to promote spontaneous regression of viral skin infection. Purified oil from coix seed has also been suggested to increase the peripheral CD4^+^ lymphocytes. We, herein, attempt to shed more light on the way through which coix seed affects the human systemic immune function by hypothesizing that a central role to these changes could be played through changes in the gut microbiota. To that end, healthy adult males (*n* = 19) were divided into two groups; 11 of them consumed cooked coix seed (160 g per day) for 7 days (intervention), while the other eight were given no intervention. One week of coix seed consumption lead to an increase of the intestinal *Faecalibacterium* abundance and of the abundance (as % presence of overall peripheral lymphocytes) of CD3^+^CD8^+^ cells, CD4^+^ cells, CD4^+^CD25^+^ cells, and naïve/memory T cell ratio. As the relationship of microbiota and skin infection has not been clarified, our findings could provide a clue to a mechanism through which coix seed could promote the spontaneous regression of viral skin infections.

## 1. Introduction

A systematic examination of the effects of traditional herbal medicines as well as a subsequent clarification of their mechanisms could allow for their effective use and provide opportunities for the development of new medicines [1,2]. Coix (*Coix lacryma-jobi* L. var. *ma-yuen* Stapf) is a grain-bearing perennial plant that belongs to the *Poaceae* family. It is widely known as adlay or pearl barley, and its grain is used for food, medicine and decoration, while its seed is used in Chinese herbal medicine [3]. Most Chinese herbal medicines are made by combining different ingredients of plant and animal sources; the coix seed, however, is mostly used on its own, as a single agent [3]. Thereby, the verification of its efficiency and the investigation of the mechanisms of its pharmacological action are expected to be rather straightforward.

A recent comprehensive review of the human trials undertaken in order to study the effects of coix seed consumption [4] has suggested that: (i) the coix seed may promote the spontaneous regression of skin viral infections [5,6], and (ii) kanglaite (a purified oil of the coix seed used in cancer treatment) increases the CD4^+^ T cell presence in the peripheral blood of cancer patients undergoing cancer therapy [7,8,9,10]. Thereby, coix seed should be rightfully considered to affect the cellular immune function. However, due to the limitations in the individual studies examined by the aforementioned review, the effects are not conclusive and the specific immune or clinical indicators upon which coix seed may act, are still unidentified [4]. Therefore, it is important to investigate the influence of coix seed on specific indices of the human immune system. It is worth noting that with regard to coixenolide [11,12,13,14] and coixol [15,16], that have both been reported as exclusive derivatives of the coix seed, their effects on immune cells have not yet been clarified. In addition, the influence of coix seed on the gut microflora has not been reported, despite the fact that roles of microbiota on health, especially, the concept that diets and herbal medicines can affect metabolism and immunity through the intestinal microbiota has been gaining attention [17,18,19,20,21,22,23]. Moreover, the relationship of microbiota and skin infection has not been clarified.

Our hypothesis was that coix seed consumption influences the systemic immune function through changes in the gut microbiota. In order to investigate this hypothesis, we examined the influence of coix seed consumption on blood and biochemical indices, percentages of peripheral lymphocyte subsets, and plasma cytokine levels along with the gut microbiota in healthy male adults. We, herein, demonstrate that one week of coix seed consumption increases intestinal *Faecalibacterium* abundance, the abundance (as % presence in overall peripheral lymphocytes) of CD3^+^CD8^+^ cells (killer T cell), CD4^+^ cells (helper T cells), CD4^+^CD25^+^ cells (regulatory T cells; Treg cells), and naïve/memory T cell ratio, while on the other hand decreases the abundance (as % presence in overall peripheral lymphocytes) of CD3^−^CD56^+^ cells (natural killer cells; NK cells) and CD3^+^CD45RA^−^CD45RO^+^ cells (memory T cells) in healthy male adults.

## 2. Materials and Methods

### 2.1. Participants

Healthy male adults were recruited from 6 November to 10 December of 2019 via the study’s website. The inclusion criteria were as follows: (i) being male aged 20–64 years old, (ii) being in good condition in order to complete the intervention safely, (iii) being a person who can understand the study’s rationale and instructions, and (iv) being willing to participate in the study voluntarily and to provide a written consent. The exclusion criteria were as follows: (i) receiving a treatment or prescribed medication from a physician, (ii) suffering from a severe cardiovascular disorder, liver dysfunction, renal dysfunction, respiratory disorder, endocrine disorder, metabolic disorder, or having a history of these, (iii) having previously experienced any allergic reaction to coix seed, (iv) being a person whose blood has been drawn (e.g., through blood donation) to a volume exceeding 200 mL within 1 month prior to the start of this study, or to a volume of 400 mL within 3 months prior to the study’s start, (v) having participated in another clinical trial within the past 3 months or being currently participating in another clinical trial, (vi) being a smoker, and (vii) habitually consuming 60 mL or more of alcohol per day.

Eligible candidates complying with all of the inclusion criteria and not meeting any of the exclusion criteria were thoroughly informed about the purpose, the methodology and the possible adverse events of the study, as well as about the policies in place in order to protect their rights as participants. Written consent was obtained from all participants.

Nineteen participants were randomly allocated to either the coix seed consumption group (CS; *n* = 11) or the control group (CN; *n* = 8) by using the Research Randomizer (https://www.randomizer.org/) (accessed on 11 December, 2019). One participant belonging to the CS group dropped out before the intervention because he failed to attend the first visit. The remaining 18 participants completed the study and were included in the analysis (Figure 1). Their average age and anthropometric characteristics did not differ when compared between the CS and the CN group (Table 1).

This study was approved by the Ethics Committee of the Juntendo University Graduate School of Health and Sports Science (approval number: 31–72), and it was performed in compliance with the ethical standards of the 1964 Declaration of Helsinki and its later amendments or comparable ethical standards. This study was registered in the UMIN Clinical Trials Registry (ID: UMIN000038831) prior to start of any intervention.

### 2.2. Study Design

Diet influences the intestinal microbiota in a relatively short period of time. David et al. have studied the influence of a plant-based diet and of an animal-based diet for five consecutive days on intestinal microbiota, and were able to find a significant change induced by the “animal-based diet” within 1 day after the diet reached the colon, and a return to baseline within 2 days after returning to the normal diet [24]. Wilck et al. have reported that a 6 g per day consumption of sodium chloride for 14 days has managed to decrease intestinal *Lactobacillus* and increase peripheral IL-17-producing helper T cells (Th17) as well as blood pressure in 12 healthy men [25]. In the same report, an experiment on mice revealed that the changes in Th17 cells in the intrinsic mucosal layer of the small intestine, the spleen and the spinal cord, occurred within 3 days after the changes affecting the *Lactobacillus* in the intestine took place [25]. Based on these studies, the duration of the intervention (coix seed consumption) in this study was set as 7 days.

The participants finished dinner by 21:00 on 13 December 2019, and visited the Tamura Clinic (Tsukuba, Japan) at 08:00 on 14 December 2019 (Day 1) after overnight fasting (except for water). The participants submitted their feces that were collected after 12 December 2019, and their height, weight and blood pressure were measured. Subsequently, their blood samples were collected, and the participants responded to a questionnaire about their lifestyle (including diet). This questionnaire included habitual physical activity and the dietary survey described below. For physical activity asked were the amount of time they spent doing high-intensity exercise (METs = 8.0), moderate-intensity exercise (METs = 4.0), and walking (METs = 2.5) during a habitual week. From the responses, the amount of habitual weekly exercise (METs×h/week) was determined.

Like on Day 1, the participants finished dinner by 21:00 on 20 December 2019, and visited the Tamura Clinic (Tsukuba, Japan) at 08:00 on 21 December 2019 (Day 8) after overnight fasting (except for water). The participants delivered their diaries, had their height, weight and blood pressure measured again, and had their blood sampled. Samples of their first feces after Day 8 were also submitted.

### 2.3. Dietary Survey

The participants’ habitual diet was assessed through a brief, self-administered, diet-history questionnaire (BDHQ); a confirmed and valid questionnaire for Japanese people that is used to assess the habitual intake of energy-adjusted nutrients [26]. The answer of one participant belonging to the CS group was incomplete because the calculated energy was found to be less than a half of the estimated energy intake for physical activity level I [27], and as such, it was removed from the analysis. The remaining 17 participants’ dietary habits were analyzed.

### 2.4. Hematological and Biochemical Indices

Serum total protein, albumin (ALB), albumin/globulin ratio (A/G), total-bilirubin (T-Bill), aspartate transaminase (AST), alanine transaminase (ALT), gamma-glutamyltranspeptidase (GGT), triglyceride (TG), high density lipoprotein-cholesterol (HDL-C), low density lipoprotein-cholesterol (LDL-C), uric acid and creatinine (CRE) were analyzed by using a clinical chemistry analyzer, namely BioMajesty JCA-BM8600 (JEOL, Tokyo, Japan). On the other hand, the automated hematological analyzer Sysmex NX-9000 (Sysmex, Kobe, Japan) was used for the analyses of the following hematological parameters: white blood cell count, red blood cell count, hemoglobin (Hb), hematocrit (Ht), mean corpuscular volume (MCV), mean corpuscular hemoglobin (MCH), MCH concentration and platelet (PLT) count. All analyses were conducted by using standard procedures at the Kotobiken Medical Laboratories (Tokyo, Japan).

### 2.5. Intestinal Microbiota

Each participant collected his feces by using the disposable paper device Rakuryu Cup (Takahashi Keisei, Yamagata, Japan). Approximately 1 g of feces was collected in a tube containing zirconium dioxide beads (BioMedical Science, Tokyo, Japan) and 2 mL of RNAlater (Thermo Fisher Scientific, Tokyo, Japan). With the lid closed tightly, the tube was shaken vigorously about 40 times, and was then stored at 4 °C until DNA extraction.

A 150 µL aliquot of the mixed feces was added to 850 µL of TE buffer containing RNase (to a final concentration of 100 µg/mL; Invitrogen, Tokyo, Japan) and lysozyme (to a final concentration of 3.0 mg/mL; Sigma-Aldrich Japan, Tokyo, Japan), and was subsequently mixed and shaken mildly for 1 h at 37 °C. The purified achromopeptidase was then added (to a final concentration of 2000 U/mL; Fuji Film Wako, Osaka, Japan) and the mixed solution was shaken for 30 min at 37 °C. Following that, sodium dodecyl sulfate and proteinase K were added (to a final concentration of 1% and 1 mg/mL, respectively; Nacalai Tesque, Kyoto, Japan) and the mixture was incubated for 1 h at 55 °C. DNA was extracted with the use of phenol:chloroform:isoamyl alcohol (at a 25:24:1 ratio), precipitated by isopropanol, washed with 75% ethanol, and then dissolved in 200 µL of TE buffer.

The V1–V2 region of the 16S rRNA gene was amplified by PCR with primers 27Fmod 5′-AGRGTTTGATYMTGGCTCAG-3′ and 338R 5′-TGCTGCCTCCCGTAGGAGT-3′. The obtained amplicons (~330 bp) were purified using the AMPure XP (Beckman Coulter, Tokyo, Japan). The amount of DNA was quantified by a TBS-380 Mini-Fluorometer (Turner Biosystems, Sunnyvale, CA, USA) by using the Quant-iT Picogreen dsDNA assay kit (Thermo Fisher Scientific, Waltham, MA, USA).

The 16S metagenomic sequencing was conducted by MiSeq (Illumina, San Diego, CA, USA) according to the manufacturer’s protocol. The paired-end reads were merged based on the overlapping sequences identified by the fastq-join program. Reads with an average quality value of <25 and inexact matches to both universal primers, were excluded.

After removing the primer sequences, the remaining reads were rearranged according to the quality value. The selected 3000 reads were clustered into OTUs with a 97% pairwise-identity cutoff, by using the UCLUST program (Edgar 2010) version 5.2.32 (https://www.drive5.com) (accessed on 6 February 2000). The taxonomic assignment of each read was made by a similarity searching against the Ribosomal Data Project and the National Center for Biotechnology Information genome database, by using the GLSEARCH program. For the calculation of the microbial abundance, the taxa with a relative abundance of >0.1% were considered as positive.

The DNA extraction, amplification, sequencing, and taxonomic assignment were conducted at MyMetagenome (Tokyo, Japan).

One participant of the CS group failed to collect his feces at the end of the study (post-intervention; Post). Thereby, the pre-intervention (Pre) data of this participant were excluded from the analysis. Therefore, the numbers of the analyzed participants were nine for the CS and eight for the CN group.

### 2.6. Peripheral Lymphocyte Subsets

The peripheral lymphocyte subsets were analyzed by using a BD FACSVia Flow Cytometer (BD Biosciences, San Jose, CA, USA) with the following antibodies against surface markers: FITC mouse IgG1 κ isotype control, FITC mouse anti-human CD3, FITC mouse anti-human CD4, FITC mouse anti-human CD8, FITC mouse anti-human CD45RA, PE mouse IgG1 κ isotype control, PE mouse anti-human CD25, PE mouse anti-human CD56, PE mouse anti-human CD4, PE mouse anti-human CD45RO, PE-Cy^TM^5 mouse IgG1 κ isotype control, and PE-Cy^TM^5 mouse anti-human CD3. The analyses were conducted at a private clinical laboratory (IMUH, Tokyo, Japan).

### 2.7. Plasma Cytokines

A total of 12 plasma cytokines were analyzed by using the LEGENDplex™ Human Th Cytokine Panel (BioLegend, San Diego, CA, USA). This panel quantified 12 human cytokines—namely, interleukin-2 (IL-2), IL-4, IL-5, IL-6, IL-9, IL-10, IL-13, IL-17A, IL-17F, IL-22, interferon-gamma (IFN-γ) and tumor necrosis factor-alpha (TNF-α)—that were collectively secreted by Th1, Th2, Th9, Th17, and Th22 cells. The analyses were conducted at a private clinical laboratory (IMUH, Tokyo, Japan).

### 2.8. Statistical Analysis

The MicrobiomeAnalyst (https://www.microbiomeanalyst.ca) (accessed on 10 September 2020) was used for the analysis of the intestinal microbiota and the dietary survey. Diversity of microbiota was calculated in OUT level. On the other hand, the hematological and biochemical indices, as well as the peripheral lymphocyte subsets, were analyzed by using the MetaboAnalyst 5.0 (https://www.metaboanalyst.ca/) (accessed on 10 September 2020). The statistical softwares SPSS version 23 (IBM Japan) and R ver. 3.6.1 were used for the other statistical analyses. Statistical significance was set at *p* < 0.05 and at |R| > 0.4 for the Spearman’s correlation coefficient.

## 3. Results

### 3.1. Dietary Habit and Health Condition

The participants’ dietary habits were assessed by the comparison of BDHQ between the CS and the CN group. The observed mean (SD) of energy intake and macronutrient ratio of CS group was 2640 kcal/day (980 kcal/day), protein:fat:carbohydrate (P:F:C) = 16.4% (3.3%):29.3% (7.3%):54.4 (10.5%), and the CN group was 1989 kcal/day (536 kcal/day), P:F:C = 13.9% (3.1%):27.4% (14.1%):58.7% (16.8%), with no significant difference between groups. The intakes of total weight, water as well as 45 nutrients and food substances were expressed as density (per 1000 kcal), and subsequently, the diversity was compared between the CS and the CN group. The α-diversity did not differ in any indices analyzed: Observed (*p* = 0.686), Chao1 (*p* = 0.492), ACE (*p* = 0.767), and Shannon (*p* = 0.154). The β-diversity was assessed by the Bray-Curtis index, only to show that there was no difference between the CS and the CN group (PERMANOVA, *p* = 0.686). Therefore, no significant difference regarding the dietary habit was observed between the two groups. The mean (SD) of the habitual physical activity of the participants was 62.7 METs×h/week (120.4 METs×h/week) and 51.0 METs×h/week (38.3 METs×h/week), the CS and the CN group, respectively; there observed no statistical difference. Moreover, according to the diary, all the participants allocated to the CS group consumed the given coix seed completely, while the health status of all participants was reported as uneventful throughout the intervention period.

### 3.2. Hematological and Biochemical Indices

The CS group participants demonstrated lower PLT levels (than those of the CN ones) before the intervention (Pre) (*p* = 0.049). The within subject change between Pre and Post revealed significant decreases in the CRE of the CS group (*p* = 0.003) and in the MCV of the CN group (*p* = 0.041). However, both parameters’ levels were all within reference ranges (Appendix A).

### 3.3. Gut Microbiota

The gut bacterial composition was analyzed by the 16s rRNA-based large-scale genome sequencing, yielding an average of 38,842 raw reads per sample, of which 20,000 reads were filtered and an average of 1852.5 reads passed the filter (Appendix A). The filtered 3000 reads were used for OTU analysis, and an average of 4.5 phyla per sample (Figure 2A; Appendix A) and a total of 166 genera (average of 46.9 per sample; Figure 2B; Appendix A) were identified.

The changes in the α-diversity of OUT (ACE, Chao1, Observed, and Shannon indices) were analyzed. The CS intake was found decreased by Chao1 (*p* = 0.028) and ACE (*p* = 0.014), while the CS intervention provided no significant changes (Figure 3). No significant changes were recorded in the Observed and Shannon indices for both the CS and the CN group (Appendix A). Changes in the β-diversity of OUT were also analyzed and presented with no statistical significance (Appendix A).

In the case of the CS group, the linear discriminant analysis effect size (LEfSe) between Pre and Post revealed significant increases in the *Faecalibacterium prausnitzii* (Pre: 218.00; Post: 312.89; LDA = 1.69; *p* = 0.022) at the species level and in the *Faecalibacterium* (Pre: 218.33; Post: 312.89; LDA = 1.69; *p* = 0.022) at the genus level, whereas no significant changes were observed at any taxonomic level in the CN group (Appendix A). Since the *Faecalibacterium prausnitzii* is the only species classified in the genus *Faecalibacterium*, the latter was subjected to further analysis.

The mean abundance of *Faecalibacterium* in the CS group was found increased (from 7.7% to 10.9%; *p* = 0.028), with the increase being evident in eight out of nine participants. On the other hand, no significant change was observed in the CN group. Thereby, the abundance observed in the CS group became significantly greater than that in the CN group after the intervention (Post; *p* = 0.013), while it did not differ at all before the intervention (Pre; *p* = 0.236) (Figure 4).

### 3.4. Peripheral Lymphocyte Subsets

Coix seed consumption resulted into a significant increase in the killer T cell (CD3^+^CD8^+^; *p* = 0.002), helper T cell (CD4^+^; *p* = 0.049), and regulatory T cell (CD4^+^CD25^+^; *p* = 0.020) percentages, as well as in the naïve/memory T cell ratio (*p* = 0.020), while at the same time it resulted into a significant decrease in the natural killer T cell (CD3^−^CD56^+^; *p* = 0.038) and memory T cell (CD3^+^CD45RA^−^CD45RO^+^; *p* = 0.005) percentages. On the other hand, a significant change was only observed in the case of the killer T cell (CD3^+^CD8^+^) percentages (*p* = 0.025) in the CN group (Figure 5; Appendix A).

The abundance of *Faecalibacterium* in the CS group after the intervention was correlated with the CD3^+^CD8^+^ (R = −0.450) and the CD3^+^CD56^+^ (R = −0.517) percentages (Appendix A). On the other hand, there was no significant correlation (|R| > 0.4) between any of the lymphocyte subsets and *Faecalibacterium* in the overall studied population, including controls and pre-intervention (Appendix A).

In the CS group, the observed changes (Post-Pre) in the *Faecalibacterium* abundance were positively correlated with the change (Post-Pre) in the CD3^−^CD56^+^ percentage (R = 0.700) and negatively with the change (Post-Pre) in the CD3^+^CD45RA^+^CD45RO^−^ (R = −0.750) and in the CD3^+^CD45RA^−^CD45RO^+^ (R = −0.450) percentages. In contrast, changes in the CD3^−^CD56^+^ percentage were found to be negatively correlated (R = −0.643) with changes in the *Faecalibacterium* abundance in the CN group (Appendix A).

### 3.5. Plasma Cytokines

A significant decrease in the plasma TNF-α levels was observed in both the CS and the CN groups. The levels of IL-17F were found significantly lower in the CS than in the CN group, in both Pre and Post instances (Appendix A).

The correlation coefficients between the observed plasma cytokine changes (Post/Pre) and the respective changes in relative peripheral lymphocyte subset and gut *Faecalibacterium* abundance were examined. In the CS group, changes in gut *Faecalibacterium* abundance were correlated with changes in IL-9 (R = 0.533) and IFN-γ (R = 0.667) levels. Changes in IL-9 levels were correlated with the CD3^−^CD56^+^ (R = 0.491) and the CD3^+^CD45RA^−^CD45RO^+^ (R = −0.636) percentages. Moreover, changes in the IFN-γ levels were correlated with the CD3^−^CD56^+^ (R = 0.455) and the CD3^+^CD45RA^+^CD45RO^−^ (R = −0.770) percentages. Meanwhile, the above relationships were not consistent in the case of the CN group. Other significant correlations (|R| > 0.4) were observed, but there was no consistent relationship between the CS and the CN group (Appendix A).

## 4. Discussion

The one-week consumption of coix seed provoked changes in gut microbiota and systemic immunity. The gut *Faecalibacterium* abundance increased. Within the same time period, the relative peripheral CD3^+^CD8^+^ (killer T cell), CD4^+^ (helper T cell), and CD4^+^CD25^+^ (Treg cell) percentages, as well as the naïve/memory T cell ratio increased, and the relative peripheral CD3^−^CD56^+^ (NK cell) and CD3^+^CD45RA^−^CD45RO^+^ (memory T cell) percentages decreased (Figure 6).

The increase in the relative peripheral blood CD4^+^ percentage has been reported in several meta-analyses as an effect of kanglaite; an injectable emulsion of purified oil preparation of coix seed [7,8,9,10]. Kanglaite is used in China with the aim of improving the outcome and the quality of life of cancer patients undergoing cancer treatment (e.g., chemotherapy, radiotherapy, and/or surgery). The meta-analyses have focused on cancer patients undergoing cancer therapy that impairs the immune system [7,8,9,10]. Therefore, it is not clear whether the observed increase in the relative peripheral CD4^+^ percentage is the result of a direct effect on the immune system or a reduction in the damage to the immune system caused by the cancer therapy. In our study, the relative peripheral CD4^+^ and CD4^+^CD25^+^ percentages increased in seven and eight out of ten participants that consumed coix seed, respectively. The results indicate that coix seed consumption increases helper T cells, especially Treg, availability in healthy individuals.

We, herein, observed an increase in the percentage of relative CD4^+^ and CD3^+^CD8^+^ percentages in peripheral lymphocytes. Thereby, a relative decrease in the other subsets should be expected. However, no significant decrease in the relative CD3^+^CD56^+^ (NK T cell), CD8^+^CD56^+^ (CD8^+^ NK cell), and CD3^+^CD45RA^+^CD45RO^−^ (naïve T cell) percentages was observed. This result indicates that these cells are also relatively increased. On the other hand, the relative CD3^−^CD56^+^ (NK cell) percentage was found significantly decreased. However, the decrease in circulation does not indicate the decrease in the systemic NK cell function, because NK cells are mobilized in circulation by a sympathetic stimulation (such as exercise) and migrate out of the circulation by cytokines (such as IL-6) [28]. A decrease in the relative CD3^+^CD45RA^−^CD45RO^+^ (memory T cell) percentage was also observed, but this should only be a relative decrease and not indicate a functional suppression. Therefore, coix seed consumption is suggested to activate both CD4^+^ and CD8^+^ cells, which are central to cellular immunity, as well as NK T cells and CD8^+^ NK cells.

In Japan, the coix seed is used as an ethical drug to treat verrucae planae juveniles (plane warts) and verruca vulgaris (common warts) [29,30], both being viral infections of the skin [31]. These are localized infections that usually regress spontaneously without medical intervention [31]. Fushiki et al. have also reported that the coix seed extract can accelerate the spontaneous regression of human papillomavirus infection [6]. Moreover, a multicenter, randomized, double blind, placebo-controlled, parallel-group study on molluscum contagiosum has also shown that coix seed consumption might promote spontaneous regression [5]. Human papillomaviruses infect skin epithelial cells, although they do not induce a lysis of the infected cells [32]. Thereby, viral antigens have smaller chance to encounter any antigen-presenting cells (such as dendritic cells), and due to that, immune responses are less likely to occur [32]. In this study, the peripheral CD4^+^, the CD8^+^, the NK T and the CD8^+^ NK cells were suggested to increase as a result of the consumption of coix seed. These cells are involved in the removal of virus-infected cells and therefore, the activation of these immune cells could account for the reported enhanced regression of viral infections as a result of coix seed consumption.

In the present study, we hypothesized that the coix seed consumption affects the immune function via the gut microbiota and examined the latter as well as the relative peripheral lymphocyte subset abundance. Coix seed consumption was found to significantly decrease the α-diversity of fecal microbiota (ACE & Chao1) and to significantly increase gut *Faecalibacterium* abundance. Since no change was observed with regard to the CN group, the aforementioned changes can be attributed to the coix seed consumption.

*Faecalibacterium* is a Gram-positive, anaerobic bacterium found in the human intestinal flora that ferments dietary fiber in order to produce butyric acid and other short-chain fatty acids [33]. *Faecalibacterium* is reported to decrease in the intestinal microbiota of Crohn’s disease patients [34,35]. Butyric acid is known to exert anti-inflammatory effects on the intestine by inhibiting the mobilization and the pro-inflammatory activity of neutrophils, macrophages, dendritic cells, and effector T cells, and by increasing the number and activity of regulatory T cells [36]. Therefore, a decrease in the gut *Faecalibacterium* abundance may be linked to Crohn’s disease, an inflammatory bowel disease, through the decrease of butyric acid in the intestine. Meanwhile, in patients with psoriasis, intestinal *Faecalibacterium* has been reported to be increased, while *Bacteroides* has been reported to be decreased [36]. Although the cause and pathogenesis of psoriasis has not yet been clarified, it is thought to be an immune system disorder that causes the skin to regenerate faster than normal [37]. Therefore, *Faecalibacterium* could affect not only the intestinal tract but also the systemic immune system. In this study, an increase in gut *Faecalibacterium* abundance and changes in the relative peripheral lymphocyte subset percentages were simultaneously observed following the consumption of coix seed. The results could indicate that coix seed consumption influences the systemic immune system via the intestinal *Faecalibacterium*.

The relative percentages of the peripheral CD3^+^CD8^+^ (R = −0.450) and CD3^+^CD56^+^ (R = −0.517) cells after coix seed consumption were significantly correlated with the gut *Faecalibacterium* abundance; on the other hand, the relative peripheral CD3^+^CD8^+^ cell percentage was significantly increased while the relative CD3^+^CD56^+^ cell percentage did not change after coix seed consumption. In the overall data (including those of the CN group), the relative percentages of the peripheral lymphocyte subsets that changed significantly, did not significantly correlate with the abundance of gut *Faecalibacterium*. These results indicate that the abundance of gut *Faecalibacterium* is not directly related to the relative percentage of a specific peripheral lymphocyte subset.

When comparing changes (Post-Pre) with regard to the gut *Faecalibacterium* abundance and the relative percentage of lymphocyte subset changes before and after coix seed intake, the gut *Faecalibacterium* in the CS group correlated positively with the CD3^−^CD56^+^ (R = 0.700) and negatively with the CD3^+^CD45RA^+^CD45RO^−^ (R = −0.750) and the CD3^+^CD45RA^−^CD45RO^+^ (R = −0.450) relative percentages. Thereby, changes in the gut *Faecalibacterium* abundance could provoke the relative change of these subsets. However, since no congruent relationship was observed regarding the CN group, we cannot simply assume that only *Faecalibacterium* changes were involved. In order to clarify the relationship between the gut *Faecalibacterium* abundance and the relative peripheral lymphocyte subset distribution, it is necessary to elucidate the implicated mechanism(s) including factors that mediate both.

In order to investigate the mechanism by which the gut *Faecalibacterium* abundance affects the relative peripheral lymphocyte subset expression, 12 plasma cytokines secreted collectively by helper T cells (Th1, Th2, Th9, Th17, and Th22) were analyzed by the LEGENDplex™ Human Th Cytokine Panel (BioLegend, San Diego, CA, USA). A significant decrease in the TNF-α levels was observed in the CS group, but this change was not specific to coix seed consumption, as it was also observed in the CN group.

We examined the association between changes in plasma cytokine levels (Post/Pre) and changes in gut *Faecalibacterium* abundance and relative peripheral lymphocyte subset percentages (Post-Pre), but we were unable to find any consistent association in either the CS or the CN group. These results suggest that a specific cytokine (if such exists) that is affected by the gut *Faecalibacterium* and affects the relative percentages of the peripheral lymphocyte subsets is not present among the 12 examined plasma cytokines. Therefore, this study demonstrates that coix seed consumption is associated with changes in gut *Faecalibacterium* abundance and the relative peripheral lymphocyte subset percentages, but the mechanisms that exist between the two, including whether there is a direct link between them, have remained unelucidated.

Our study suffers from some limitations. Firstly, it is necessary to ensure that the dietary habits of the subjects do not produce bias, since diet directly affects the gut microbiota. In this study, healthy male adults residing in the same area were recruited, and their dietary habits were assessed by BDHQ and revealed no significant difference between the CS and the CN group. However, we did not control for the diet during the intervention period, thereby the possibility cannot be excluded that several other dietary factors during the intervention might have confounded the obtained results. The effects of coix seed consumption could be more clearly observed by conducting the same trial under controlled diet. Secondly, the relative percentages of the peripheral lymphocyte subsets were measured, but the actual lymphocyte count was not determined. Thereby, we found an increase in the relative CD4^+^ cell percentage, but we could not clearly discuss whether the number of these cells in circulation was increased or whether the numbers of other cell subsets in circulation were decreased. It is necessary to measure the lymphocyte count as well as the relative percentage of the peripheral lymphocyte subsets. In addition, this study was conducted with a limited number of male participants. Future research should be undertaken by recruiting a larger number of participants, including females.

## 5. Conclusions

In this study, we were able to demonstrate that a week of coix seed consumption increases gut *Faecalibacterium* abundance and the relative percentages of peripheral CD3^+^CD8^+^ (killer T cell), CD4^+^ (helper T cells), CD4^+^CD25^+^ (Treg cells) and naïve/memory T cell ratio and decreases the relative percentages of CD3^−^CD56^+^ (NK cells) and CD3^+^CD45RA^−^CD45RO^+^ (memory T cells) lymphocytes. Our findings are expected to be a clue to the mechanism through which coix seed consumption could exert the previously reported spontaneous regression of viral skin infections.

## Figures and Tables

**Figure 1 nutrients-13-04079-f001:**
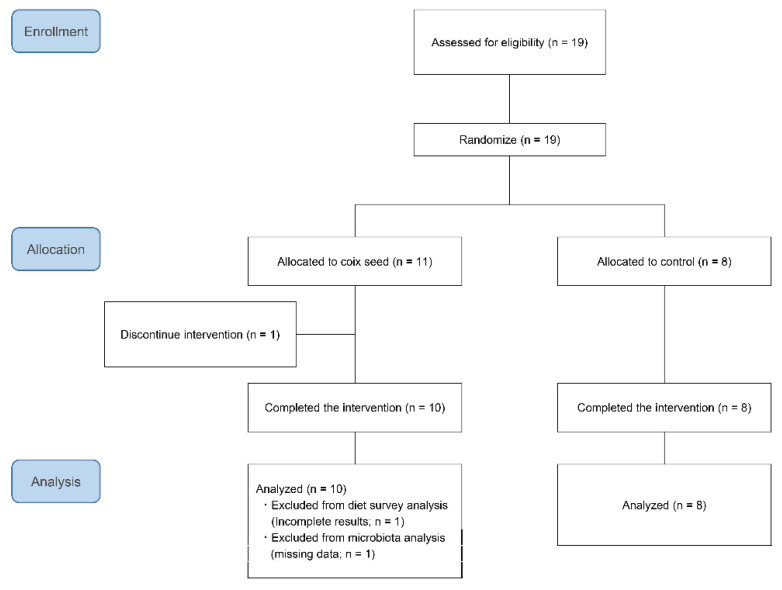
Diagrammatic synopsis of the participants’ enrollment and allocation, as well as of the analysis phase of the study.

**Figure 2 nutrients-13-04079-f002:**
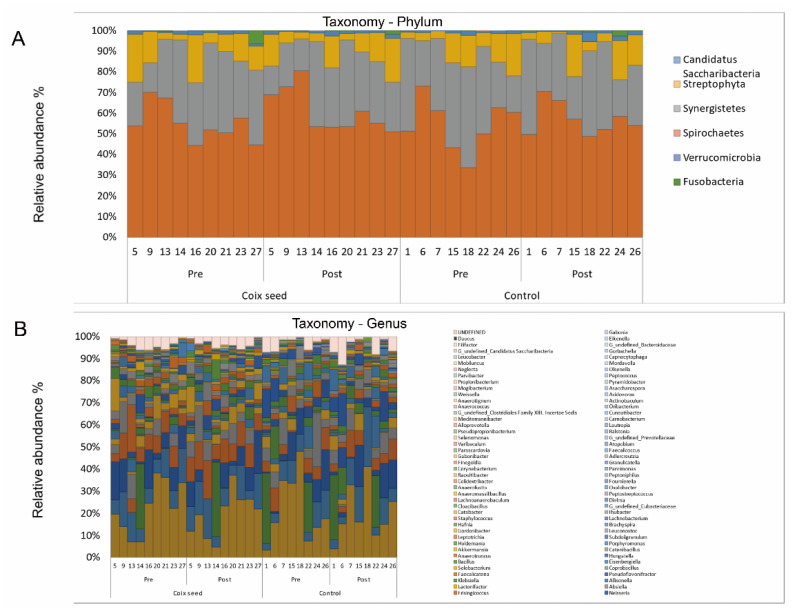
Phyla (**A**) and genera (**B**) of the gut bacteria identified in each sample of feces, before and after the consumption of coix seed for 7 days.

**Figure 3 nutrients-13-04079-f003:**
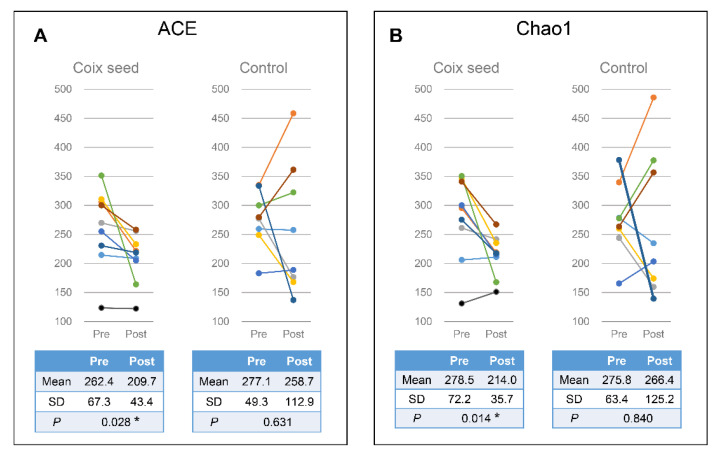
Alpha-diversity in the participants’ gut microbiota following either the ACE (**A**) or the Chao1 (**B**) analysis. The coix seed (CS) group showed significant decrease in both ACE (**A**) and Chao1 (**B**), whereas no change was observed in the control (CN) group.

**Figure 4 nutrients-13-04079-f004:**
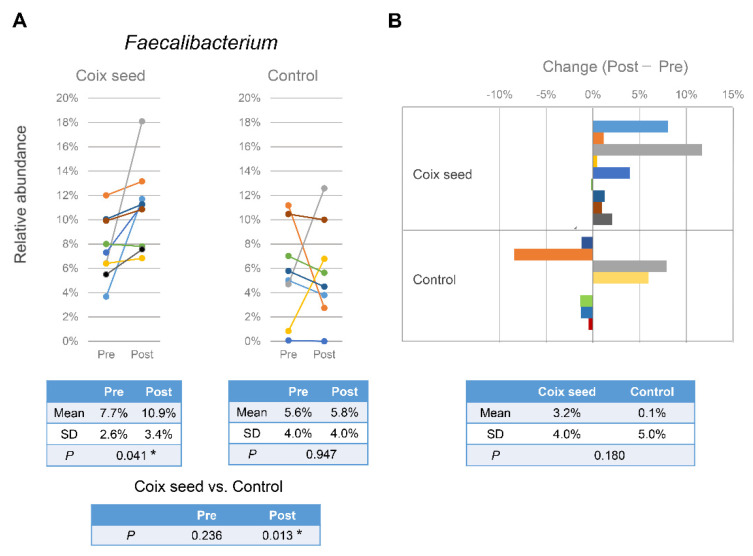
Intestinal *Faecalibacterium* relative abundance as identified in each participant before and after the intervention (**A**), and the recorded change between the post- and the pre-intervention (Post -Pre) samples in each participant as a percentage (**B**).

**Figure 5 nutrients-13-04079-f005:**
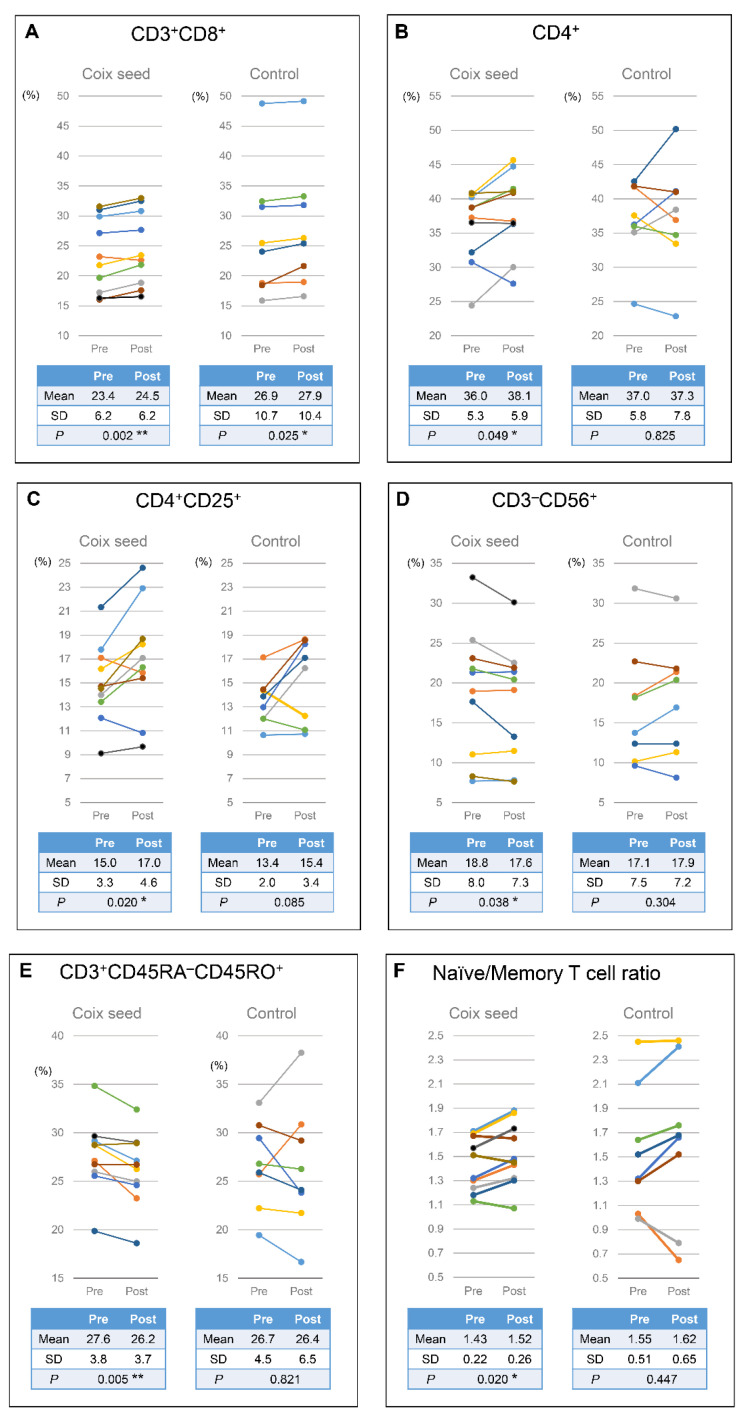
Relative changes in the peripheral lymphocyte subsets as identified in each participant before and after (Pre and Post) the intervention; (**A**) killer T cells (CD3^+^CD8^+^); (**B**) helper T cells (CD4^+^); (**C**) regulatory T cell (CD4^+^CD25^+^); (**D**) natural killer T cells (CD3^−^CD56^+^); (**E**) memory T cells (CD3^+^CD45RA^−^CD45RO^+^); and (**F**) the naïve/memory T cell ratio. After one week of coix seed consumption, the relative peripheral CD3^+^CD8^+^ (killer T cell), CD4^+^ (helper T cell), and CD4^+^CD25^+^ (Treg cell) percentages, as well as the naïve/memory T cell ratio increased, and the relative peripheral CD3^−^CD56^+^ (NK cell) and CD3^+^CD45RA^−^CD45RO^+^ (memory T cell) percentages decreased in coix seed (CS) group. In contrast, significant change was only seen in the relative percentage of killer T cells (CD3^+^CD8^+^) in control (CN) group.

**Figure 6 nutrients-13-04079-f006:**
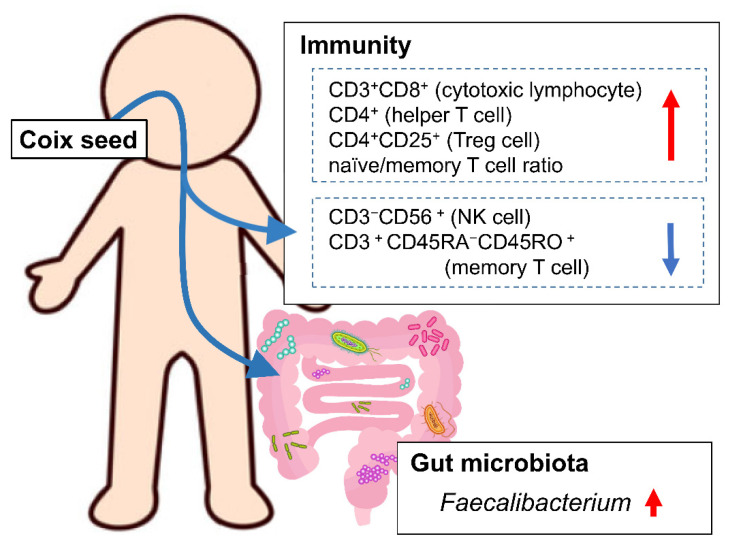
Summary of the results. The one-week consumption of coix seed increased the gut *Faecalibacterium* abundance, the relative peripheral CD3^+^CD8^+^ (killer T cell), CD4^+^ (helper T cell), and CD4^+^CD25^+^ (Treg cell) percentages, as well as the naïve/memory T cell ratio, and decreased the relative peripheral CD3^−^CD56^+^ (NK cell) and CD3^+^CD45RA^−^CD45RO^+^ (memory T cell) percentages.

**Table 1 nutrients-13-04079-t001:** Synopsis of the age and the anthropometric characteristics of the study’s participants, as assigned to the two groups.

	**Coix Seed** **Consumption Group (CS)**			**Control** **Group (CN)**			** *p* **
	n	Mean	*SD*	*n*	Mean	*SD*	
**Age** (years)	10	27.3	9.5	8	28.1	9.9	0.860
**Height** (cm)	10	173.2	6.6	8	171.7	3.8	0.544
**Body weight** (kg)	10	73.2	15.5	8	71.4	8.6	0.766
**BMI** (kg/m^2^)	10	24.3	4.0	8	24.2	2.3	0.954

*p*: *p* value of the significance deriving from the statistical analysis performed via a Welch’s *t*-test.

## Data Availability

The data presented in this study are available on request from the corresponding author. The data are not publicly available elsewhere due to ethical restrictions.

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
