# Peer review of "Coix Seed Consumption Affects the Gut Microbiota and the Peripheral Lymphocyte Subset Profiles of Healthy Male Adults"

_nutrients, 2021, doi:10.3390/nu13114079_

Round 1
Reviewer 1 Report
Dear Authors,
Firstly, I would like to congratulate you on an attempt to discuss such an important topic and to present the findings. I sincerely hope that this comments/suggestion assist in the improvements of the manuscript.
Major:
- The manuscript contains number of grammatical and academic expression errors as well as some typographical errors that will require to be corrected. In saying this, the approaches used for the methodological aspect are very interesting and appropriate. Congratulations to the team I have thoroughly enjoyed reading your work.
- I would suggest that authors also include nutritional intake analysis in their results sections in particular macronutrient breakdown of the diet at start and at the end of the treatment for both groups. Any changes in diet should also reflect the changes in gut, inflammatory markers as well.
- Did authors collect any physical exercise data as this can be also considered as influential capacity of the observed (not/observed) findings.
- The structure of the introduction section is somewhat non-organized and it would be beneficial to include more “logical flow” of the hypothesis development. For example, authors are drawing most of the information from only one review (REF 4). The problem with this is that comprehensive reviews only summarize the studies and I suggest that for this type of article, original studies are used with clear indications towards the developed hypothesis. Furthermore, the introduction section will require ‘back-up’ with references for several statements.
Minor:
- Authors should consider consistent referencing.
- Figure 1 and 2 will require higher resolution in particular Figure 2B
Reviewer 2 Report
The current manuscript is interesting. However, in order to publish, the authors need to improve some items as below:
-The relationship between skin infection and gut microbiota/immunity is not clear. It should be explained briefly in the abstract and introduction.
-According to the findings, the conclusion of the abstract also needs to modify.
-The introductory part of the discussion section is to be rewritten in order to make it clearer.
- A schematic diagram of the summary is recommended and cited in the conclusion section as a figure.
